# Understanding Data Influence in Reinforcement Finetuning

**Haoru Tan**[1]    **Xiuzhe Wu**[5][†]    **Sitong Wu**[3]    **Shaofeng Zhang**[4]

**Yanfeng Chen**[2]    **Xingwu Sun**[2]    **Jeanne Shen**[5]    **Xiaojuan Qi**[1][†]

[1]The University of Hong Kong    [2]Hunyuan Team, Tencent

[3]The Chinese University of Hong Kong    [4]University of Science and Technology of China

[5]Stanford University

`hrtan@eee.hku.hk, stone-wu@link.cuhk.edu.hk`

## Abstract

Reinforcement fine-tuning (RFT) is essential for enhancing the reasoning and generalization capabilities of large language models, but its success heavily relies on the quality of the training data. While data selection has been extensively studied in supervised learning, its role in reinforcement learning, particularly during the RFT stage, remains largely underexplored. In this work, we introduce RFT-Inf, the first influence estimator designed for data in reinforcement learning. RFT-Inf quantifies the importance of each training example by measuring how its removal affects the final training reward, offering a direct estimate of its contribution to model learning. To ensure scalability, we propose a first-order approximation of the RFT-Inf score by backtracking through the optimization process and applying temporal differentiation to the sample-wise influence term, along with a first-order Taylor approximation to adjacent time steps. This yields a lightweight, gradient-based estimator that evaluates the alignment between an individual sample's gradient and the average gradient direction of all training samples, where a higher degree of alignment implies greater training utility. Extensive experiments demonstrate that RFT-Inf consistently improves reward performance and accelerates convergence in reinforcement fine-tuning.

## 1   Introduction

Reinforcement Fine-Tuning (RFT) [1–4] has emerged as a powerful technique for refining the capabilities of large language models (LLMs) [1, 2, 5, 3] by leveraging reward-driven optimization [6, 7]. Unlike Supervised Fine-Tuning (SFT) [8–10], which guides model behavior through direct supervision [11], RFT enables models to learn more robust reasoning strategies and generalize beyond seen data [1, 11, 12]. Recent studies have confirmed the critical role of RFT in advancing the reasoning capabilities of LLMs, particularly in complex tasks requiring multi-step inference and decision-making.

**Background.** A fundamental aspect of successful RFT lies in the construction and utilization of high-quality training data [1, 13, 14]. However, current understanding of data's role in RFT remains

---

[†]Corresponding Authors

39th Conference on Neural Information Processing Systems (NeurIPS 2025).

largely heuristic or empirical, often guided by reward trends observed during training [13, 15] or the perceived difficulty of training examples [16, 17]. Despite its importance, there remains a lack of principled, quantitative methods for assessing the impact of individual samples on RFT outcomes. Addressing this gap is essential for designing more efficient and effective training pipelines.

**Our Method.** In this work, we propose RFT-Inf for quantifying the influence of the individual training sample in reinforcement fine-tuning. The core idea behind RFT-Inf is to measure how the removal of a single sample influences the final training reward, providing a direct estimate of its contribution to the model's learning process. While a brute-force approach would require re-training the model after removing each sample, such computation is prohibitively expensive. To address this, we introduce a scalable, first-order approximation of the RFT-Inf score derived from backtracking through the optimization process and applying temporal differentiation to the sample-wise influence term, along with a first-order Taylor approximation to adjacent time steps. This results in a lightweight gradient-based estimator with linear complexities that measures the alignment between a sample's gradient and the average gradient direction of all training samples across different training stages, see Eq.(5). A higher alignment corresponds to a higher RFT-Inf score, indicating greater sample importance. Intuitively, if a sample's gradient is closely aligned with the overall optimization direction, it is more representative and beneficial to training. Removing such a sample would degrade performance, highlighting its importance. We further provide a theoretical worst-case error bound for our first-order approximation, offering insights into its reliability.

We validate RFT-Inf across multiple benchmarks, demonstrating its effectiveness in identifying high-impact samples and improving final model performance. Using RFT-Inf for data selection yields significant performance gains over baselines that rely on full datasets or heuristic selection strategies. Notably, in mathematical reasoning tasks, we only require about **20%** data selected by our data influence estimator to achieve more stable training and superior results compared to using the entire dataset. Compared to various heuristic or rule-based data selection methods, our approach significantly outperforms them in both performance and generalization.

## 2 Related Work

**Reinforcement Fine-Tuning.** Recent breakthroughs in large language models (LLMs) [2, 1, 3] have been significantly driven by Reinforcement Fine-Tuning (RFT) [6, 7, 1, 18, 19], a technique that refines LLM behavior using reward-guided optimization. Unlike Supervised Fine-Tuning (SFT) [8–10], which aligns model outputs with labeled responses [11], RFT employs reinforcement learning to adapt models based on feedback signals. Typically, RFT relies on rule-based reward functions and reinforcement learning algorithms such as Proximal Policy Optimization (PPO) [6]. For instance, DeepSeek-R1 [1] adopts Group Relative Policy Optimization (GRPO) [7], using binary rewards to indicate answer correctness in tasks like mathematics [20] and coding [21], achieving impressive performance. Several studies suggest that RFT enhances cognitive abilities such as reflection and self-correction [22, 1], while also improving generalization across tasks [11]. Much of the current RFT research focuses on algorithmic improvements. For example, VinePPO [23] addresses limitations in PPO's value networks for complex reasoning tasks by introducing unbiased Monte Carlo estimates for better credit assignment. Other works aim to simplify GRPO, e.g., by removing the KL-divergence term to achieve more robust empirical results [18, 24].

Most recently, some emerging research explores data-centric approaches to improve RFT [13, 14, 18, 15]. For example, LIMR [13] selects samples by analyzing changes in reward trends, while the Historical Variance Score [15] prioritizes data with high reward variability, suggesting that such samples are more impactful. Other works incorporate sample difficulty as a selection criterion to strike a balance between learning efficiency and model robustness [16, 17]. Despite these advances, existing work on data selection for RFT is largely heuristic and lacks principled methods for quantifying the influence of individual training samples. As a result, most methods rely on empirical intuition rather than theoretical insight. This highlights an urgent need for data-centric frameworks that enable quantitative analysis of sample importance during RFT, ultimately guiding more effective training.

**Data Influence Analysis.** Analyzing the impact of individual training samples is a longstanding problem in machine learning [25, 26]. The canonical approach involves removing a sample and re-training the model to observe the effect—an idea that is precise but computationally prohibitive [26–28]. To overcome this, many works propose theoretical estimators that approximate sample

influence without full retraining. A seminal method by Koh and Liang [26] estimates sample influence using the inverse Hessian-gradient product, assuming the model parameters change smoothly under small data perturbations. Building on this, subsequent work has improved the efficiency of Hessian computations through decomposition techniques [29–32], incorporated group-level effects [33, 34], or integrated influence estimation with training procedures [35, 36].

However, these approaches are predominantly designed for supervised learning and often depend on key assumptions: a well-defined, twice-differentiable objective function, and convexity of the loss landscape [26, 37, 38]. Such conditions facilitate tractable approximation but do not hold in reinforcement learning, where the optimization objective is to maximize expected reward rather than minimize a loss. Consequently, applying these influence functions directly to reinforcement learning is challenging. Moreover, data influence research in supervised learning has been leveraged for tasks such as data selection [26, 39–43] and attribution [29, 44–49], but similar progress in reinforcement learning remains limited. This disconnect motivates the development of new, scalable influence estimation tools tailored specifically for RFT scenarios.

# 3 Preliminaries

This section introduces the foundational concepts of reinforcement fine-tuning (RFT) [2, 1, 14, 7, 50, 3] as applied to large language models (LLMs) [51, 52]. We consider a training dataset $\mathcal{Z} = z_i = (s_i, y_i)_{i=1}^N$ composed of structured question–answer pairs, where each sample $z_i$ consists of a prompt or question $s_i$ and a corresponding deterministic ground-truth answer $y_i$, e.g., a math problem and its correct solution. Let $\pi_\theta$ denote the policy model parameterized by $\theta$, representing the LLM. In this formulation, the input state $s$ corresponds to the prompt, and the action $a$ is the model's generated response. A scalar reward $r$ measures the quality of the response $a$ for the given prompt $s$. The objective of RFT is to fine-tune the LLM using reinforcement learning to produce correct and high-quality answers [6, 7]. A commonly adopted reward scheme directly compares the model output with the ground truth [1], assigning a reward of $r = 1$ if the output matches the correct answer and $r = -1$ otherwise.

**Objective Function.** The standard RFT objective is to optimize the policy $\pi_\theta$ by maximizing the expected advantage:

$$\mathcal{J}(\theta) = \mathbb{E}_{(s,a)}\Big[A(s,a)\Big], \tag{1}$$

where the advantage function is defined as $A(s,a) = r(s,a) - v$, representing the relative merit of taking action $a$ in state $s$ compared to a baseline value $v$. Optimization proceeds via stochastic gradient ascent: $\theta_{t+1} = \theta_t + \eta_t \nabla_\theta J(\theta_t)$, where $\eta_t$ is the learning rate and the gradient of $\mathcal{J}(\theta)$ is derived as: $\nabla_\theta \mathcal{J}(\theta) = \mathbb{E}_{(s,a)}\Big[A(s,a) \cdot \nabla_\theta \log \pi_\theta(a|s)\Big]$.

Modern RFT implementations typically enhance this vanilla formulation to improve stability and performance. We now briefly review two widely used variants: Proximal Policy Optimization (PPO) [6] and Group Relative Policy Optimization (GRPO) [7, 1], which serve as the foundation for many current RFT advancements [18, 12, 50, 53].

**Proximal Policy Optimization** (PPO) [6] introduces a clipping mechanism to stabilize training by constraining policy updates. Its objective is formulated as:

$$\mathcal{J}^{\text{PPO}}(\theta) = \mathbb{E}_{(s,a)}\left[\min\left(\frac{\pi_\theta(a|s)}{\pi_{\theta_{\text{old}}}(a|s)}A(s,a), \text{clip}\left(\frac{\pi_\theta(a|s)}{\pi_{\theta_{\text{old}}}(a|s)}, 1-\epsilon, 1+\epsilon\right)A(s,a)\right)\right], \tag{2}$$

where $\epsilon$ is a clipping hyperparameter. $\pi_\theta(a|s)$ is the probability of taking action $a$ in state $s$ according to the current policy with parameters $\theta$, and $\pi_{\theta_{\text{old}}}(a|s)$ is the probability of taking action $a$ in state $s$ according to the old policy with parameters $\theta_{\text{old}}$.

**Group Relative Policy Optimization** (GRPO) [7, 1] extends PPO by defining advantages through relative comparisons within a group of sampled responses for the same prompt. For each prompt $s$, a group of outputs $a_1, \ldots, a_G$ is sampled from the old policy $\pi_{\theta_{\text{old}}}$. The GRPO objective is:

$$\mathcal{J}^{\text{GRPO}}(\theta) =$$

$$\mathbb{E}_{(s,a_i)}\left[\frac{1}{G}\sum_{i=1}^G \min\left(\frac{\pi_\theta(a_i|s)}{\pi_{\theta_{\text{old}}}(a_i|s)}A(s,a_i), \text{clip}\left(\frac{\pi_\theta(a_i|s)}{\pi_{\theta_{\text{old}}}(a_i|s)}, 1-\epsilon, 1+\epsilon\right)A(s,a_i)\right) - \beta\text{KL}\left(\pi_\theta \| \pi_{\text{ref}}\right)\right], \tag{3}$$

where $\epsilon$ is the clipping hyperparameter and $\beta$ is the coefficient controlling the weight of the KL-divergence regularization to ensure the optimized model does not deviate excessively from the reference model (*e.g.*, the initial policy model); and $A(s, a_i) = \frac{r_i - \text{mean}(r_1, \ldots, r_G)}{\text{std}(r_1, \ldots, r_G)}$ is the advantage, where $r_i$ corresponds to the reward of each output. GRPO eliminates the need for a separate value network, reducing computational cost and simplifying training. For more details, we refer readers to the original GRPO papers [7, 1].

## 4 Method

In this section, we introduce our proposed method. We begin by defining the data influence estimator and presenting a first-order approximation for efficient computation, along with a theoretical analysis of the approximation error. We then describe how this score can be used to select high-impact samples to improve the performance of RFT.

### 4.1 Data Influence in Reinforcement Finetuning

We propose a principled approach to assess the importance of each training sample by measuring its influence on the final performance of reinforcement fine-tuning through evaluating how model performance changes when the sample is removed from training.

**Definition.** Let $\mathcal{Z}$ denote the training dataset consisting of samples $z = (s, y)$, where $s$ is a prompt and $y$ is its deterministic answer (e.g., a math problem and its unique solution). Let $\mathcal{Z}/z$ denote the dataset excluding sample $z$, and let $\theta^*_{\mathcal{Z}}$ and $\theta^*_{\mathcal{Z}/z}$ denote the parameters obtained by training on $\mathcal{Z}$ and $\mathcal{Z}/z$, respectively. Assuming a reward function that assigns $r = 1$ if the model output matches the ground truth and $r = -1$ otherwise [1, 14], we define the data influence estimator of a sample $z$ as:

$$\mathcal{D}(z) = \mathcal{J}\left(\theta^*_{\mathcal{Z}}\right) - \mathcal{J}\left(\theta^*_{\mathcal{Z}/z}\right), \tag{4}$$

where $\mathcal{J}(\cdot)$ denotes the expected reward. This score is directional; it not only quantifies the importance of a sample but also distinguishes whether the sample is beneficial or detrimental to training. A large positive value of $\mathcal{D}(z)$ indicates that removing the sample significantly reduces the final reward, suggesting that the sample is beneficial. Conversely, a negative value implies that the sample is harmful, as removing it would increase the reward. Values near zero indicate negligible influence. Thus, this metric provides a direct and interpretable measure of each sample's contribution to RFT.

**Efficient First-order Approximation.** Precisely Direct computation of $\mathcal{D}(z)$ is impractical, as it requires retraining the model for each sample. To address this issue, we introduce a first-order approximation of the RFT-Inf score by backtracking the score defined in Eq. (4) through the optimization process. We apply temporal differentiation and a first-order Taylor approximation to adjacent time steps. The detailed derivation is provided in the supplementary material. Below is the approximation with linear complexities:

$$\hat{\mathcal{D}}(z) = \sum_t \frac{2\eta_t}{N} \left\langle \mathcal{G}_z^{(t)}, \mathcal{G}_{\mathcal{Z}}^{(t)} \right\rangle, \tag{5}$$

where $\left\langle :, : \right\rangle$ is the inner-product operator, $\mathcal{Z}/z$ represents the dataset $\mathcal{Z}$ excluding sample $z : (s, y)$, and $\eta_t$ is the learning rate at time step $t$. $T$ and $N$ denote the maximum number of time steps and the training set size, respectively. $\mathcal{G}_z^{(t)}$ represents the policy gradient contributed by sample $z : (s, y)$ at time step $t$. Typically, for most RL algorithms, we have $\mathcal{G}_z^{(t)} = \hat{A}(s, a) \nabla \log \pi_{\theta_t}(a|s)$, where $\hat{A}(s, a)$ is the advantage function. In the case of GRPO [7], $\mathcal{G}_z^{(t)} = \sum_i^G \hat{A}(s, a_i) \nabla \log \pi_{\theta_t}(a_i|s)$, where $G$ is the group size. The other gradient term $\mathcal{G}_{\mathcal{Z}}^{(t)}$ is defined as the gradient over all samples, that is, $\mathcal{G}_{\mathcal{Z}}^{(t)} = \sum_{z \in \mathcal{Z}} \mathcal{G}_z^{(t)}$.

This approximation evaluates the alignment between a sample's gradient and the overall training direction. A higher value indicates that the sample's influence aligns with the global training trajectory, marking it as representative and valuable. This can be interpreted as follows: if the sample's gradient $\mathcal{G}_z^{(t)}$ exhibits a high degree of alignment with the average gradient vector $\mathcal{G}_{\mathcal{Z}}^{(t)}$ for all time steps, then

**Algorithm 1:** Data Selection Pipeline

---

**Require:** A dataset $\mathcal{Z} = \{(s_i, y_i)\}_{i=1}^N$ and selection budget $\delta$; A large language model $\pi_\theta$ and a reinforcement fine-tuning algorithm RFT

1: Train the model $\pi_\theta$ for $E$ epochs on $\mathcal{Z}$ using RFT and save checkpoints:
   $\{\theta^1, \ldots, \theta^E\} \leftarrow \text{RFT}(\pi_\theta, \mathcal{Z}, E)$
2: **for** each sample $z_i = (s_i, y_i) \in \mathcal{Z}$ **do**
3:    Calculate the data influence estimator $\hat{\mathcal{D}}(z_i)$ according to Eq. (5)
4: **end for**
5: Select the top-$\delta$ samples based on their data influence estimators to form the new subset $\mathcal{Z}_{\text{new}}$
6: **return** The subset model $\mathcal{Z}_{\text{new}}$

---

optimizing the network using sample $z$ will have a similar effect optimizing with the full training set. This indicates that the sample $z$ is a representative and important sample. Therefore, this sample is regarded as a more representative and significant one.

### 4.2  Error Analysis

First, we provide a worst-case error bound for the proposed approximation, demonstrating its robustness under some mild assumptions:

**Proposition 1.**  *Under the assumptions that the log-likelihood function $\log \pi_\theta(a|s)$ exhibits $\ell$-Lipschitz continuity and that the adventure value is upper-bounded by $A_{max}$, the approximation error of Eq. (5) is bounded as follows:*

$$\left| \mathcal{D}(z) - \hat{\mathcal{D}}(z) \right| \leq \mathcal{O}\left( \left[ \frac{\eta_{\max}(4N+4)}{N}(\ell A_{\max})^2 + 2\eta \ell^2 A_{\max} \right] T \right), \qquad (6)$$

*where $T$ denotes the maximum number of iterations and $N$ is the number of all training data.*

The above proposition formalizes the relationship between the approximation error and influencing factors, particularly the training duration $T$. To mitigate errors in practice, we adopt a strategy of early stopping during the surrogate training phase, avoiding full convergence. Specifically, we limit the number of update epochs to a small value (e.g., two epochs) to ensure the effective execution of surrogate training while also controlling the theoretical maximum error. Remarkably, the resulting error bound remains tighter than many known bounds for influence functions in supervised learning [41, 54], despite the added challenges posed by the reinforcement learning setting.

### 4.3  Complexity Analysis

Then, we discuss the computational complexity of our estimator. Eq. (5) offers an efficient approximation of the data influence estimator with linear computational complexity. Further, rather than summing over all training steps, we adopt a Monte Carlo sampling strategy [55, 56], selecting a small number of evenly spaced time steps to estimate the full summation since checkpoints from nearby iterations will be similar. Let the number of selected checkpoints be $C$. By employing this strategy, the total computational complexity for scoring the entire dataset is $\mathcal{O}\left(NE + NC\right)$, where $NE$ is the cost of surrogate training, and $N$ represents the number of training samples and $C$ denotes the number of checkpoints (which also corresponds to the number of sampled checkpoints, as we select one checkpoint for each epoch). This represents a substantial efficiency gain over the naive approach, which requires retraining the model for each sample and incurs a cost of $\mathcal{O}\left((N-1)E\right)$ per sample, resulting in $\mathcal{O}\left(N(N-1)E\right)$ overall. Leveraging practical implementations such as PyTorch's efficient gradient operations [57], or computing only final-layer gradients [55, 58], further enhances scalability.

### 4.4  Implementation Details

We now describe how to use the proposed data influence estimator to identify and select high-impact samples for reinforcement fine-tuning. The full pipeline is outlined in Algorithm 1. We are given a

structured dataset $\mathcal{Z} = \{z_i = (s_i, y_i)\}_{i=1}^N$ consisting of question–answer pairs, where each sample $z_i$ comprises a prompt or question $s_i$ and a corresponding deterministic answer $y_i$. We are also provided with a pre-trained large language model, serving as the policy model $\pi_\theta$, and a reward evaluation mechanism. A common and effective approach to reward calculation involves directly comparing the generated answer to the ground truth $y$; for instance, a reward of $r = 1$ is assigned if the generated answer matches the ground truth, and $r = -1$ otherwise.

As shown in Algorithm 1: the policy model first undergoes a small number of reinforcement fine-tuning (RFT) epochs, namely $E$ epochs, on the full training dataset. We refer to this training phase as surrogate training. The choice of reinforcement learning algorithm is flexible; for instance, one may use PPO [6], GRPO [7], or any other compatible method. At the end of each epoch, a model checkpoint is saved for later use. To reduce training costs, several practical strategies can be employed. These include setting a small number of epochs (e.g., $E = 2$), or applying parameter-efficient fine-tuning techniques such as LoRA [59] instead of full model fine-tuning, as recommended by recent studies [60, 31]. These techniques have been empirically shown to significantly reduce computational overhead without negatively impacting performance. After surrogate training, the saved checkpoints are used to compute per-sample gradients as well as the overall gradient across the dataset. The data influence estimator for each sample is then computed according to Eq. (5). Using the computed data influence estimators, we perform data selection based on a predefined budget $\delta$, which specifies the size of the target subset. The top-$\delta$ samples with the highest data influence estimators are selected to form a refined training set, denoted as $\mathcal{Z}_{\text{new}}$. This curated subset is then used for the formal reinforcement fine-tuning (RFT) phase to obtain the final model parameters $\pi_\theta^*$.

## 5 Experiments

We conducted comprehensive experiments to evaluate the effectiveness of our proposed method. Sec. 5.1 is the main experiment. Then, Sec. 5.2 provides detailed ablation studies to analyze the impact of key components in our approach.

### 5.1 Main Experiments

**Models, Datasets, and Benchmarks.** We utilized the dataset released by DeepScaleR [14], which is a comprehensive mathematical dataset compiled from multiple sources, with duplicates removed and data cleaned. This dataset includes AIME problems from 1984 to 2023 and AMC problems before 2023, along with questions from the Omni-MATH [61] and STILL [62] datasets, featuring problems from various national and international mathematics competitions.

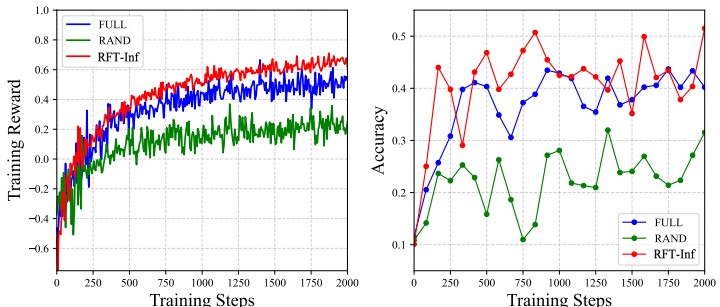

Figure 1: The reward and accuracy curve during learning on the subset selected by RFT-Inf (20% selection ratio). This experiment is conducted on DeepSeek-R1-Distill-Qwen-1.5B, and the selected benchmark is AIME24 [20].

This training dataset contains approximately 40,000 math problem-answer pairs. To evaluate the reasoning abilities of the models, we utilize five different mathematics benchmarks: AIME24 [20], MATH-500 [63], AMC23 [64], Minerva [65], and OlympiadBench [66]. Our experiments encompass a variety of model configurations, including DeepSeek-R1-Distill-Qwen-1.5B [1], DeepSeek-R1-Distill-Qwen-7B [1], and Llama-3.2-3B-Instruct [67]. Unless otherwise specified, the default algorithm used is GRPO, with a group size set to 8.

**Setups.** The experiments were conducted using the PyTorch framework on two high-performance computing servers, each equipped with eight NVIDIA H200 GPUs. For our approach, we made the following settings: during the surrogate training phase, we performed LoRA training with a LoRA

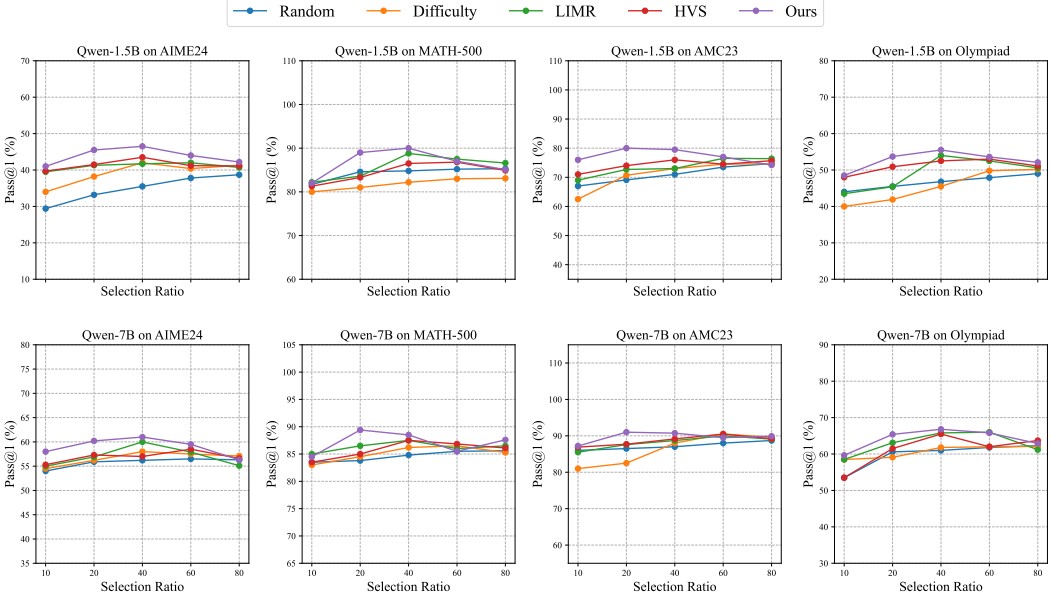

Figure 2: This figure presents the zero-shot pass@1 performance (%) of our proposed data selection method ("Ours") against several baselines (Random, Difficulty-based, LIMR, HVS) across four mathematical reasoning benchmarks (AIME24, MATH-500, AMC23, Olympiad) and two model sizes: Qwen-1.5B (DeepSeek-R1-Distill-Qwen-1.5B) and Qwen-7B (DeepSeek-R1-Distill-Qwen-7B). The performance is plotted as a function of the data Selection Ratio (from 10% to 80%), representing the budget constraint.

rank set to 16 and a total of 2 training epochs. We optimized the network using the AdamW optimizer with a constant learning rate of $1 \times e^{-6}$ and a weight decay of 0.1.

**Baselines.** We compare our method against several widely used baselines: (1) Random selection, where a subset of data is chosen uniformly at random. (2) Difficulty-driven selection, where the difficulty of each sample is estimated based on model performance. Specifically, we run inference for each sample with multiple models (DeepSeek-R1-Distill-Qwen-1.5B/7B/32B [1]), and use the average pass rate as a proxy for difficulty [17]. Based on this, we select those samples with median difficulty [68]. (3) LIMR [13], which determines sample importance by measuring the alignment between a sample's reward and the overall reward trend during training. (4) Historical Variance Score (HSV) [15], which quantifies sample importance by computing the variance in its reward values over the course of training.

To evaluate our method, we selected two models: DeepSeek-R1-Distill-Qwen-1.5B [1] and DeepSeek-R1-Distill-Qwen-7B [1]. Here, the model we use for scoring is consistent with the final trained model. Both models are derived from the Qwen series and have been distilled using DeepSeek-R1, demonstrating a solid foundational reasoning ability. We summarize the relevant experimental results in Fig. 2. Clearly, our method outperforms various baseline approaches under the most constrained budgets, particularly at very low selection ratios. Specifically, at the 20% selection ratio setting: for the 1.5B model type, our method achieves an accuracy of 45.5% in AIME24, outperforming all baseline methods, including LIMR [13] and HVS [15]. On MATH-500, our score of 89.0% is notably higher than the best baseline score of 86.2% from Full Data. In the case of the 20% selection ratio and the 7B model, our method further excels, achieving a score of 60.2% on AIME24, which is significantly higher than the next best baseline score of 57.3% from HVS [15].

## 5.2 Further study

In this subsection, we explore several key factors, including the number of checkpoints and the training method (either LoRA rank or full-parameter training), also along with the case study.

**Number of Checkpoints.** We employed a Monte Carlo strategy when approximating the estimator in Eq. (5), considering checkpoints from only a few time steps. In practice, we observed that as the

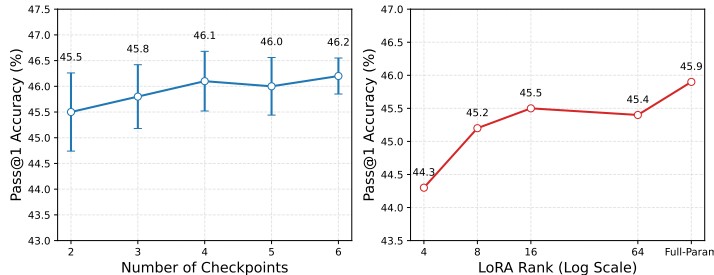

Figure 3: Ablative experimental results with DeepSeek-R1-Distill-Qwen-1.5B as the model and AIME24 as the benchmark. The performance metric is zero-shot Pass@1 Accuracy. (a) Pass@1 accuracy versus the number of checkpoints used, showing that increasing checkpoints stabilizes the training (decreasing standard deviation). (b) Pass@1 accuracy versus LoRA rank (log scale), demonstrating that performance generally improves with higher LoRA ranks, approaching the Full-Parameter result.

number of checkpoints increased, the final performance improved, see Fig. 3. This is understandable, as it effectively increases the sampling steps, thereby enhancing approximation accuracy. However, this also results in a significant decrease in scoring speed, and the improvement shows a trend of stagnation. Therefore, we opted for a relatively small number of checkpoints, specifically two, with each corresponding to one epoch.

**LoRA-Rank.** Our method requires training a surrogate model before scoring, using LoRA training with a rank of 16. LoRA (Low-Rank Adaptation) is a technique that efficiently fine-tunes large language models by incorporating low-rank decomposition into the weight updates, enabling effective adaptation with fewer parameters and reduced computational cost. This means that when approximating the estimator in Eq.(5), we do not utilize all model parameters. We tested the impact of using LoRA training and the size of the LoRA rank on final performance, see the results in Fig. 3. In conclusion, we found that using LoRA significantly improves efficiency compared to full-parameter training, without negatively impacting the performance of the data selected based on the final data influence estimator. Additionally, we discovered that even with a LoRA rank of 4, the final performance remains quite satisfactory.

**Case study.** In the following, we provide an illustrative examination of several cases, showing their assigned RFT-Inf scores and the resulting importance rankings (in descending order). For benchmarking, we include the data importance rankings generated by the alternative HVS [15] strategy. To establish an objective metric for task complexity, we use the pass rate, measured by the number of correct outputs from eight QwQ-32B [5] samples. Lower pass-rates correlate with higher question difficulty. Our analysis reveals a crucial distinction in data valuation: A high degree of concordance is found between RFT-Inf and HVS rankings for data points associated with extreme difficulty levels. For example, in both Case 1 and Case 4, which represent the spectrum's edges, both methods assign significantly low importance ranks, implying a general agreement that the marginal utility of such data is negligible. Crucially, the two ranking strategies exhibit substantial disparity when evaluating data points of intermediate or moderate difficulty. This disagreement in identifying the most valuable samples within the 'middle ground' of difficulty constitutes the fundamental driver behind the varying empirical performance observed in different data selection paradigms.

---

**Case 1**

Problem: Each vertex of a regular octagon is independently colored either red or blue with equal probability. The probability that the octagon can then be rotated so that all of the blue vertices end up at positions where there were originally red vertices is $\frac{m}{n}$, where $m$ and $n$ are relatively prime positive integers. What is $m + n$?

```
Pass-rate-by-QwQ [5]: 0/8;
Rank-by-RFT-DA: 40245;
Rank-by-HVS [15]: 39076.
```

> **Case 2**
>
> Problem: Among the 900 residents of Aimeville, there are 195 who own a diamond ring, 367 who own a set of golf clubs, and 562 who own a garden spade. In addition, each of the 900 residents owns a bag of candy hearts. There are 437 residents who own exactly two of these things, and 234 residents who own exactly three of these things. Find the number of residents of Aimeville who own all four of these things.
>
> ```
> Pass-rate-by-QwQ [5]: 5/8;
> Rank-by-RFT-DA: 3205;
> Rank-by-HVS [15]: 27855.
> ```

> **Case 3**
>
> Problem: Let $ABC$ be a triangle inscribed in circle $\omega$. Let the tangents to $\omega$ at $B$ and $C$ intersect at point $D$, and let $\overline{AD}$ intersect $\omega$ at $P$. If $AB = 5$, $BC = 9$, and $AC = 10$, $AP$ can be written as the form $\frac{m}{n}$, where $m$ and $n$ are relatively prime integers. Find $m + n$.
>
> ```
> Pass-rate-by-QwQ [5]: 4/8;
> Rank-by-RFT-DA: 1129;
> Rank-by-HVS [15]: 30724.
> ```

> **Case 4**
>
> Problem: Jen enters a lottery by picking 4 distinct numbers from $S = \{1, 2, 3, \cdots, 9, 10\}$. 4 numbers are randomly chosen from $S$. She wins a prize if at least two of her numbers were 2 of the randomly chosen numbers, and wins the grand prize if all four of her numbers were the randomly chosen numbers. The probability of her winning the grand prize given that she won a prize is $\frac{m}{n}$ where $m$ and $n$ are relatively prime positive integers. Find $m + n$.
>
> ```
> Pass-rate-by-QwQ [5]: 8/8;
> Rank-by-RFT-DA: 39275;
> Rank-by-HVS [15]: 40236.
> ```

# 6 Conclusion

In this study, we present RFT-Inf, a novel approach for data selection in reinforcement fine-tuning, aimed at finding those beneficial training examples while eliminating noisy or harmful ones. Central to RFT-Inf is a sample-level influence analysis that assesses the significance of each training example by evaluating how its exclusion impacts the final training reward, providing a clear indication of its contribution to model learning. To enhance scalability, we introduce a lightweight, gradient-based estimator with theoretical guarantees. Comprehensive experiments demonstrate that RFT-Inf consistently improves reward performance and accelerates convergence in reinforcement fine-tuning.

**Limitations.** While our method demonstrates strong performance across benchmarks, the experimental scale validated in this paper is still limited. In real-world scenarios, both the data and model sizes are significantly larger. In the future, we plan to secure additional computational resources to validate performance on a larger experimental scale.

# 7 Acknowledge

This work has been supported by the National Key R&D Program of China (Grant No. 2022YFB3608300), Hong Kong Research Grant Council - Early Career Scheme (Grant No. 27209621), General Research Fund Scheme (Grant No. 17202422, 17212923, 17215025), Themebased Research (Grant No. T45-701/22-R), and Shenzhen Science and Technology Innovation Commission (SGDX20220530111405040). Part of the described research work is conducted in the JC STEM Lab of Robotics for Soft Materials funded by The Hong Kong Jockey Club Charities Trust.

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

# A Impact Statement

This paper introduces a novel influence analysis algorithm to advance RFT training. It has potential positive societal effects, such as improving understanding of data roles in developing robust systems and possibly reducing data bias. However, there are concerns that the negative impacts could be misused for inhumane social surveillance, which legislative bodies worldwide should address seriously.

# B Mathematical Proof: Approximation

Let $\theta^t$ denote the learned parameter at the $t-$th iteration on the full dataset and $\theta^t_{-z}$ denote the learned parameter at the $t-$th iteration on the dataset without sample $z$, we use $\mathcal{D}^t(z)$ to denote the empirical expected reward at the $t$-th step,

$$\mathcal{D}^t(z) = \mathcal{J}(\theta^t) - \mathcal{J}(\theta^t_{-z}),$$

where $t \leq T$, and $\mathcal{D}(z) = \mathcal{D}^T(z)$. Note that the network on the full set and that on the subset $\mathcal{Z}/z$ started from the same initialization, hence, $\mathcal{M}^0(z) = 0$. Let's start with the identical equation below,

$$\mathcal{D}(z) = \Big(\mathcal{D}(z) - \mathcal{D}^{T-1}(z)\Big) + \Big(\mathcal{D}^{T-1}(z) - \mathcal{D}^{T-2}(z)\Big) + ... + \Big(\mathcal{D}^1(z) - \mathcal{D}^0(z)\Big) + \mathcal{D}^0(z)$$

$$= \Big(\mathcal{D}(z) - \mathcal{D}^{T-1}(z)\Big) + \Big(\mathcal{D}^{T-1}(z) - \mathcal{D}^{T-2}(z)\Big) + ... + \Big(\mathcal{D}^1(z) - \mathcal{D}^0(z)\Big) \qquad (7)$$

$$= \Delta\mathcal{D}^T + \Delta\mathcal{D}^{T-1} + ... + \Delta\mathcal{D}^1.$$

Let's take one single item $\Delta\mathcal{D}^t$ as an example,

$$\Delta\mathcal{D}^t = \mathcal{D}^t(z) - \mathcal{D}^{t-1}(z) = \Big[\mathcal{J}(\theta^t) - \mathcal{J}(\theta^{t-1})\Big] - \Big[\mathcal{J}(\theta^t_{-z}) - \mathcal{J}(\theta^{t-1}_{-z})\Big] \qquad (8)$$

By using the first-order Taylor approximation and according to the gradient-ascent algorithm in RFT, we have that,

$$\mathcal{J}(\theta^t) - \mathcal{J}(\theta^{t-1}) \approx \nabla\mathcal{J}(\theta^{t-1})\Big(\theta^t - \theta^{t-1}\Big) = \eta^t ||\nabla\mathcal{J}(\theta^{t-1})||^2,$$

where the policy gradient $\nabla\mathcal{J}(\theta^{t-1}) = E_{(s,a)}(A(a,s)\nabla\log\pi_{\theta^t}(a|s))$. Hence, we have the equation for the differential term $\Delta\mathcal{D}^t$, that is,

$$\Delta\mathcal{D}^t = \eta^t ||\nabla\mathcal{J}(\theta^{t-1})||^2 - \eta^t ||\nabla\mathcal{J}(\theta^{t-1}_{-z})||^2.$$

Given a specific sample $z$, we present a unified empirical expected reward formulation for RFT:

$$\mathcal{J}_\epsilon = \frac{1}{N}\sum_{s\neq z} A(a_s, s) + (\frac{1}{N} + \epsilon) \cdot A(a_z, z),$$

where $\epsilon$ is a coefficient. Hence, we have the expected reward on the full training set by setting $\epsilon = 0$ and the subset without sample $z$ by (approximately) setting $\epsilon = \frac{-1}{N}$, which is a convention in influence analysis for supervised learning [26, 33, 34, 31, 41, 58, 29, 46]. We can treat the $(\frac{1}{N} + \epsilon) \cdot A(a, z)$ as the new reward for sample $z$. Hence, the policy gradient given $\mathcal{J}_\epsilon$ is:

$$\nabla\mathcal{J}_\epsilon = \frac{1}{N}\sum_{s\neq z} A(a_s, s)\nabla\log\pi_\theta(a_s|s) + (\frac{1}{N} + \epsilon) \cdot A(a_z, z)\nabla\log\pi_\theta(a_z|z).$$

Here, we use the Taylor approximation again to approximate $\nabla\mathcal{J}(\theta^{t-1}_{-z})$ with $\nabla\mathcal{J}(\theta^{t-1})$, which is also a convention in influence analysis for supervised learning [26, 33, 34, 31, 41, 58, 29, 46],

$$\nabla\mathcal{J}(\theta^{t-1}_{-z}) \approx \nabla\mathcal{J}(\theta^{t-1}) + \frac{\partial\nabla\mathcal{J}^{t-1}_\epsilon}{\partial\epsilon}\Big|_{\epsilon=0}\Big((\epsilon = \frac{-1}{N}) - (\epsilon = 0)\Big) = \nabla\mathcal{J}(\theta^{t-1}) - \frac{1}{N}g^{t-1}_z, \quad (9)$$

where $g^{t-1}_z = \nabla\log\pi_{\theta^{t-1}}(a_z|z)A(a_z, z)$.

For other variants, the formulation for $g_z^{t-1}$ also needs to be adapted, for example, the group relative formulation for GRPO [7]:

$$g_z^{t-1} = \frac{1}{G}\sum_{i \leq G} \nabla \log \pi_{\theta^{t-1}}(a_{z,i}|z)\hat{A}(a_{z,i}, z),$$

where $a_{z,i}$ means the $i$-th answer from the model given the input question/prompt $z$, $\hat{A}(a_{z,i}, z) = \left(\frac{\pi_\theta(a_i|s)}{\pi_{\theta_{old}}(a_i|s)}A(s, a_i), \text{clip}\left(\frac{\pi_\theta(a_i|s)}{\pi_{\theta_{old}}(a_i|s)}, 1-\epsilon, 1+\epsilon\right)A(s, a_i)\right) - \frac{\beta}{G}\text{KL}\left(\pi_\theta\|\pi_{ref}\right)$.

Hence, we have,

$$\begin{aligned}
\Delta\mathcal{D}^t(z) &\approx \eta^t\|\nabla\mathcal{J}(\theta^{t-1})\|^2 - \eta^t\|\nabla\mathcal{J}(\theta_{-z}^{t-1})\|^2 \\
&\approx \eta^t\|\nabla\mathcal{J}(\theta^{t-1})\|^2 - \eta^t\left\langle\nabla\mathcal{J}(\theta^{t-1}) - \frac{1}{N}g_z^{t-1} \;,\; \nabla\mathcal{J}(\theta^{t-1}) - \frac{1}{N}g_z^{t-1}\right\rangle \\
&= \frac{2\eta^t}{N}\left\langle\nabla\mathcal{J}(\theta^{t-1}), g_z^{t-1}\right\rangle - \frac{\eta^t}{N^2}\left\langle\nabla g_z^{t-1}, g_z^{t-1}\right\rangle \\
&\approx \frac{2\eta^t}{N}\left\langle\nabla\mathcal{J}(\theta^{t-1}), g_z^{t-1}\right\rangle,
\end{aligned} \tag{10}$$

where $\left\langle :,: \right\rangle$ is the inner-product operator. By substituting the approximation for $\mathcal{D}^t(z)$ into Eq.(7), we have that,

$$\mathcal{D}(z) = \Delta\mathcal{D}^T + \Delta\mathcal{D}^{T-1} + ... + \Delta\mathcal{D}^1 \approx \sum_t \frac{2\eta^t}{N}\left\langle\nabla\mathcal{J}(\theta^t), g_z^t\right\rangle, \tag{11}$$

where $0 \leq t < T$.

## C   Mathematical Proof: Error Bound

We provide a worst-case error bound for the proposed approximation, demonstrating its robustness under some mild assumptions.

**Proposition 1.** *Under the assumptions that the log-likelihood function $\log\pi_\theta(a|s)$ exhibits $\ell$-Lipschitz continuity and that the adventure value is upper-bounded by $A_{max}$, the approximation error is bounded as follows:*

$$\left|\mathcal{D}(z) - \hat{\mathcal{D}}(z)\right| \leq \mathcal{O}\left(\left[\frac{\eta_{max}(4N+4)}{N}(\ell A_{max})^2 + 2\eta\ell^2 A_{max}\right]T\right), \tag{12}$$

*where $T$ denotes the maximum number of iterations and $N$ is the number of all training data.*

Our estimator is based on the time-domain differential operation over $\mathcal{D}(z) = \sum_t \Delta\mathcal{D}^t(z)$, and performing Taylor approximation on $\Delta\mathcal{D}^t(z)$. Hence, the overall error is bounded by

$$|\mathcal{D}(z) - \hat{\mathcal{D}}(z)| \leq T|\mathcal{D}^t(z) - \hat{\mathcal{D}}^t(z)|.$$

We set $d$ as the upper bound of the norm of the policy gradient. Hence, we have $d \leq A_{max}\ell$, where $\ell$ is the Lipschitz constant of the log-likelihood function $\log\pi_\theta(a|s)$ with $\ell$-Lipschitz continuity.

According to Eq.(10), the error comes from the following parts: The first approximation term, $\Delta\mathcal{D}^t(z) \approx \eta^t\|\nabla\mathcal{J}(\theta^{t-1})\|^2 - \eta^t\|\nabla\mathcal{J}(\theta_{-z}^{t-1})\|^2$, using the first-order Taylor approximation and the gradient-ascent algorithm in RFT, the error is bounded by

$$\begin{aligned}
&\left|\Delta\mathcal{D}^t(z) - \left[\eta^t\|\nabla\mathcal{J}(\theta^{t-1})\|^2 - \eta^t\|\nabla\mathcal{J}(\theta_{-z}^{t-1})\|^2\right]\right| \\
&= \left|\left[\mathcal{J}(\theta^t) - \mathcal{J}(\theta^{t-1})\right] - \left[\mathcal{J}(\theta_{-z}^t) - \mathcal{J}(\theta_{-z}^{t-1})\right] - \left[\eta^t\|\nabla\mathcal{J}(\theta^{t-1})\|^2 - \eta^t\|\nabla\mathcal{J}(\theta_{-z}^{t-1})\|^2\right]\right| \\
&\leq 2\ell\eta d + 2\eta d^2.
\end{aligned} \tag{13}$$

The second approximation term in approximating $\nabla \mathcal{J}(\theta_{-z}^{t-1})$ with $\nabla \mathcal{J}(\theta^{t-1})$, that is, $\nabla \mathcal{J}(\theta_{-z}^{t-1}) \approx \nabla \mathcal{J}(\theta^{t-1}) - \frac{1}{N}g_z^{t-1}$. We have

$$\left| \eta^t \|\nabla \mathcal{J}(\theta_{-z}^{t-1})\|^2 - \eta^t \|\nabla \mathcal{J}(\theta^{t-1}) - \frac{1}{N}g_z^{t-1}\|^2 \right| \leq 2\eta d^2 + \frac{2\eta}{N}d^2 + \frac{\eta}{N^2}d^2. \qquad (14)$$

The last error term comes from ignoring $\frac{\eta^t}{N^2}\left\langle \nabla g_z^{t-1}, g_z^{t-1} \right\rangle$ in the final approximation in Eq.(10), this error is bounded by $\frac{\eta}{N^2}d^2$.

Hence, the overall error is bounded by

$$\begin{aligned}
|\mathcal{D}(z) - \hat{\mathcal{D}}(z)| \leq T|\mathcal{D}^t(z) - \hat{\mathcal{D}}^t(z)| &\leq T\left(2\ell\eta d + 2\eta d^2 + 2\eta d^2 + \frac{2\eta}{N}d^2 + \frac{\eta}{N^2}d^2 + \frac{\eta}{N^2}d^2\right) \\
&= T\left((4\eta + \frac{2\eta}{N} + \frac{2\eta}{N^2})d^2 + 2\ell\eta d\right) \\
&\leq T\left((4\eta + \frac{4\eta}{N})d^2 + 2\ell\eta d\right) \\
&= T\left(\frac{\eta_{\max}(4N+4)}{N}d^2 + 2\ell\eta d\right) \\
&\leq \left[\frac{\eta_{\max}(4N+4)}{N}(\ell A_{\max})^2 + 2\eta\ell^2 A_{\max}\right]T
\end{aligned}$$

$$(15)$$

