# OpenReview forum: "Understanding Data Influence in Reinforcement Finetuning"
_NeurIPS.cc/2025/Conference — NeurIPS 2025 poster_

### Official Review · Reviewer_rGmp · 2025-06-26

**Clarity:** 3
**Significance:** 3
**Originality:** 3
**Rating:** 4
**Confidence:** 3

**Summary:**

This paper applies the TracIn method to RLF, proposing a framework called RFT-DA. It estimates the contribution of individual training samples through a first-order influence approximation with linear complexity (i.e. TracIn), and uses these scores to filter and prioritize valuable examples. This data selection approach improves training stability and efficiency. Experiments on mathematical reasoning tasks show that using only 20% of selected data achieves comparable or better performance than full-data training, with the 7B model matching or outperforming the o1-mini baseline.

**Questions:**

1. How do you choose the 90% threshold for the score summation of top-scoring samples? How sensitive are the results to this choice?
2. Can RFT-DA be used to dynamic or continually evolving datasets?

**Ethical Concerns:**

["NO or VERY MINOR ethics concerns only"]

**Final Justification:**

My concerns have been addressed. I'm willing to keep my original score, which reflects a positive recommendation for acceptance.

**Limitations:**

Yes.

**Paper Formatting Concerns:**

No formatting issue observed.

**Quality:**

3

**Strengths And Weaknesses:**

### Strengths
Despite some minor typos, the paper is overall well written and clearly structured. Applying the TracIn method to RLF is a novel combination. The proposed approach is theoretically simple and well motivated, with a clear derivation and sound design. The experimental evaluation is thorough, and the method shows promising potential for reducing training cost by enabling data-efficient RLF according to principled influence analysis.
### Weaknesses
One main motivation is to make training more efficient, but RLF-DA requires additional computation cost despite several practical strategies are applied. The author should elaborate in the main text regarding whether the benefits offered by RFT-DA justify its increased computational overhead.

---

> ### Author Rebuttal · Authors · 2025-07-31
>
> Response to Reviewer rGmp
>
> We truly value your acknowledgment of our efforts! Your time and contributions to the academic community are greatly appreciated. We are more than willing to answer any questions you might have!
>
> ---
>
> ### **Weakness-1: On the computation cost of RFT-DA**
>
>
> In the supplementary material (Table 3), we include a comparison with HVS on the MATH-500 dataset. Notably, HVS incurs virtually no additional time cost during the scoring phase. LIMR demonstrates comparable computational cost, as both rely solely on reward statistics collected during training and do not require any additional computation at inference. However, both methods exhibit significantly lower performance (Pass@1 of 83.6) compared to full-set training (86.2), limiting their practical utility in performance-critical scenarios.
>
> In contrast, our proposed approach achieves substantially higher performance: 89.0 with **RFT-DA** and 88.3 with the faster variant, **RFT-DA Fast**. Despite these gains, the total computation time—including surrogate training, scoring, and subset training—is significantly lower than full-set training (192.2 hours for RFT-DA, 134.2 hours for RFT-DA Fast vs. 277.6 hours for full-set training). These results will be added to the main paper for clarity. It’s worth noting that the scoring time is measured using 8 GPUs which can be significantly reduced by using more GPUs. Next, we will introduce a revised version of RFT-DA that reduces the scoring phase time from several hours to 32 seconds.
>
> **RFT-DA Dynamic: A Faster and More Efficient Variant:** To further reduce computation time for scoring, we introduce **RFT-DA Dynamic**, which eliminates the need for fixed model checkpoints by computing scores dynamically during surrogate training. Specifically, at iteration $t$, given a batch $B_t$ and a sample $z \in B_t$, we compute and store the inner product between the gradient of $z$ and the average gradient of the batch $B_t$. After training, we aggregate these stored inner products across all iterations in which each sample was involved to estimate its final score:
>
> $$D(z) \approx \sum_{t \in \{t^z_1, t^z_2, \ldots\}} \frac{2\eta_t}{N} \langle G_z^t, G_{B_t}^t \rangle,$$
>
> where $t^z_i$ denotes the iteration index of $i$-th time in which sample $z$ was used for training. This approach allows RFT-DA Dynamic to accumulate all scoring information on-the-fly during training, thus incurring negligible additional computation. The table below summarizes the comparison across methods in terms of scoring time using 8 GPUs and performance:
>
>
>
> Method   | $\~$  Time Cost of Scoring   |  $\~$ Math-500 Pass@1
> ---|---|---
> HVS| 4.7 sec $\approx 0$|83.3
> LIMR| 5.3 sec $\approx 0$ |83.6
> RFT-DA (ours)|63.1 Hours| **89.0**
> RFT-DA-fast (ours)|5 Hours| **88.1**
> RFT-DA-Dynamic (ours) | 32 sec | **88.3**
> Full-set training | - | 86.2
>
> While **RFT-DA Dynamic** yields slightly lower accuracy than the original RFT-DA, it achieves comparable computational efficiency to HVS and LIMR, requiring less than one minute for scoring. Importantly, it still significantly outperforms HVS, LIMR and full-set training in terms of final model performance, making it a highly effective and scalable solution for data selection under budget constraints. We will include the above in the paper.
>
>
> ---
>
>
> ### **Q1: On the Selection of Settings and Sensitivity of the Budget.**
>
>
> That's a great question. This issue has also been raised by other reviewers. Here, I've restated a unified response, which I hope will address your concerns.
>
> **How was the 90% budget chosen?** Rather than applying a fixed threshold on the RFT-DA score, which varies significantly across models and datasets, we opted for a more robust, generalizable strategy based on **score summation rate**. Specifically, we found through extensive experiments on GSM8K that selecting samples whose cumulative scores account for **90% of the total positive score sum** yields **the highest data efficiency**, defined as the ratio of performance gain to the amount of data used. Please refer to the table below for detailed results.
>
> This heuristic proved to be **consistently effective** across different tasks and model sizes, including experiments on MATH-500 and the Puzzle dataset. As a result, we adopted this 90% summation rate as a unified selection criterion throughout all experiments reported in the paper.
>
> **Sensitivity to budget selection.** We also evaluated the sensitivity of our method to different data selection budgets. The results are provided in the supplementary material (Table 2 and the table below), under the following experimental settings:
>
> - **Setting 1 (MATH-500):** We use the DeepSeek-R1-Distill-Qwen-1.5B model and evaluate Pass@1 accuracy on the MATH-500 benchmark.
> - **Setting 2 (GSM8K):** We use Qwen-2.5-1.5B-Instruct as the base model and evaluate Pass@1 accuracy on the GSM8K validation set.
> - **Setting 3 (Puzzle dataset):** We follow the Logic-R framework to generate 5,000 puzzle-style questions varying in difficulty and number of roles (2–5 roles). We then test generalization on 6–8 role puzzles using the same Qwen-2.5-1.5B-Instruct model.
>
> Across all three settings, we consistently observed that setting the summation threshold to **90%** yielded the **best trade-off between performance and data size**, and generalized well across datasets and tasks. This empirical consistency motivated our use of the 90% threshold in all main results reported in the paper.
>
>
>
> |                     |   Setting-1   |                |                |   Setting-2   |                |                |   Setting-3   |                |                |
> |---------------------|----------------|----------------|----------------|----------------|----------------|----------------|----------------|----------------|----------------
> |                     | Performance ↑  | budget δ ↓     | data efficiency ↑ | Performance ↑  | budget δ ↓     | data efficiency ↑ | Performance ↑  | budget δ ↓     | data efficiency ↑ |
> | No-training         | 82.8           | 0              | -              | 80.4           | 0              | -              | 0.1            | 0              | -              |
> | Full-data-training   | 86.2           | 40300          | 0.008%         | 94.2           | 7500           | 0.18%          | 82.7           | 5000           | 1.65%          |
> | Discard the negative | 87.9           | 39485          | 0.012%         | 96.8           | 7260           | 0.22%          | **83.5**       | 4832           | 1.72%          |
> | summation-rate = 95% | **89.2**      | 19295          | 0.033%         | **97.8**       | 6721           | 0.26%          | 83.3           | 3933           | 2.11%          |
> | summation-rate = 90% | 89.0          | 8530           | **0.072%**     | 97.4           | 1329           | **1.28%**      | 83.4           | 3805           | **2.18%**      |
> | summation-rate = 80% | 87.2          | 7024           | 0.062%         | 95.8           | 1230           | 1.25%          | 77.2           | 3619           | 2.13%          |
> | summation-rate = 70% | 80.5          | 6575           | -0.034%        | 94.3           | 1132           | 1.23%          | 64.8           | 3133           | 1.94%          |
>
>
> ---
>
>
> ### **Q2: Can RFT-DA be used on dynamic or continually evolving datasets?**
>
> This is a great suggestion, and thank you for raising it!
>
> **RFT-DA is indeed well-suited for dynamic or continually evolving datasets.** As shown in Equation 5, the RFT-DA score measures the **gradient alignment of a sample with respect to the gradient of the overall data**. This formulation naturally supports incremental updates. When the dataset changes, for example, through the addition of new samples or the removal of existing ones, we can simply **update the aggregate gradient** accordingly by incorporating the gradients of new samples or removing the contributions of those no longer present. This allows the RFT-DA scores to be **recomputed efficiently without retraining from scratch**, making it highly adaptable to evolving datasets.

---

> > ### Comment · Reviewer_rGmp · 2025-08-04
> >
> > I appreciate the authors’ effort in providing additional experimental results. My concerns have been resolved, and I will keep my original score.

---

> > > ### Author Response · Authors · 2025-08-04
> > > **Thanks for the response**
> > >
> > > Thank you for your response! If you have any further questions, please do not hesitate to let us know. We are more than happy to address any concerns you may have.

---

### Official Review · Reviewer_wKRB · 2025-06-30

**Clarity:** 3
**Significance:** 3
**Originality:** 3
**Rating:** 4
**Confidence:** 4

**Summary:**

RFT-DA is a data-selection framework that improves the efficiency and effectiveness of reinforcement fine-tuning (RFT). RFT-DA uses a lightweight, gradient-based estimator to evaluate the alignment between an individual sample's gradient and the average gradient direction of all training samples. RFT-DA's 7B model is on par with or outperforms the ol-mini on a number of benchmarks. The RFT-DA framework provides a scalable, first-order, gradient-based estimator for computing RFT-DA scores with linear computational complexity. The framework has been validated across a number of benchmarks, including MATH-500, AIME24, AMC23, and OlympiadBench.

**Questions:**

Q1. The experiments consistently select 8,530 samples (~20% of the dataset) based on a rule of removing negative-scoring samples and then taking the top 90% of the remaining score summation. While this procedure is explicit, the rationale for the "90% of score sum" threshold seems heuristic. How was this value determined, and how sensitive is the model's performance to this data selection budget?

Q2. What are the characteristics of samples that receive very high positive scores versus those that receive negative scores?

**Ethical Concerns:**

["NO or VERY MINOR ethics concerns only"]

**Limitations:**

I do not see any potential negative societal impact of their work.

**Paper Formatting Concerns:**

No formatting issue.

**Quality:**

3

**Strengths And Weaknesses:**

Strengths:

1.The paper is written well and easy to understand.
2. The idea is somewhat novel and justified solidly.
3. The selected data can generalize towards other models or RL algorithms.

Weaknesses:
1. Only considering math or code benchmarks is not convinced.
2.While results on 1.5B and 7B models are strong, it remains an open question whether the findings and the specific data selection ratios (e.g., 20%) would hold for much larger models (e.g., 70B+).
3. Many typos (e.g., line 346 'dmodel')

---

> ### Author Rebuttal · Authors · 2025-07-31
>
> Response to Reviewer wKRB:
>
> We sincerely appreciate your recognition of our work!. We are more than happy to address any questions you may have!
>
> ---
>
>
> ### **Weakness-1: Evaluation on More Benchmarks and Larger Models.**
>
>
> **Experiments on More Benchmarks.** We further conducted performance tests on GPQA and Arena-Hard! GPQA is a benchmark focused on assessing large models' expert-level knowledge comprehension and complex reasoning abilities, covering fields such as biotech and chemistry. Arena-Hard is an automatic evaluation tool for instruction-tuned LLMs designed to reflect whether the models meet human preferences.
> The model used for the experiments was DeepSeek-R1-Distill-Qwen-7B, and the datasets involved included DeepScaleR and a 50K subset of the Tulu-3-Preference dataset. The RL algorithm employed was Reinforce++, and the reward model used was Tulu-3-8B-RM. To ensure fairness, all methods were filtered to include only the top 90% based on summation rate.
>
> Notably, RFT-DA continues to maintain its leading advantage! We will update these results in the revision.
>
>
> Method | GPQA | Arena-Hard
> ---|---|---
> No training| 48.4 | 17.3
> HVS|52.8|26.9
> LIMR|51.4|24.2
> RFT-DA (ours)|**53.5**|**28.5**
> Full-set training|51.7|27.4
>
>
>
>
> **Experiments on Larger Models.** Thank you for your suggestion! We present the training results on the 32B (DeepSeek-R1-Distill-Qwen-32B) and 70B (DeepSeek-R1-Distill-Llama-70B) models. Here, we directly apply the subsets chosen in the 1.5B model experiments for RFT on these larger-scale models. The performance advantages of RFT-DA remain consistent with those observed on the 1.5B and 7B models, outperforming both HVS and LIMR, as well as the training results on the full set!
>
> Method|Math-500 Pass@1 for 32B|Math-500 Pass@1 for 70B
> ---|---|---
> No training|94.3|94.5
> HVS|93.7|94.1
> LIMR|92.9|93.2
> RFT-DA (ours)|**96.9**|**97.1**
> Full-set Training|95.1|96.5
>
>
> ---
>
>
>
> ### **Weakness-2: On the typos.**
>
> Thank you for pointing that out! We have conducted a thorough proofreading of the paper and corrected the typos in the revision.
>
> ---
>
>
> ### **Q1: On the Selection of Settings and Sensitivity of the Budget.**
>
>
> That's a great question.
>
> **How was the 90% budget chosen?** Rather than applying a fixed threshold on the RFT-DA score, which varies significantly across models and datasets, we opted for a more robust, generalizable strategy based on **score summation rate**. Specifically, we found through extensive experiments on GSM8K that selecting samples whose cumulative scores account for **90% of the total positive score sum** yields **the highest data efficiency**, defined as the ratio of performance gain to the amount of data used. Please refer to the table below for detailed results.
>
> This heuristic proved to be **consistently effective** across different tasks and model sizes, including experiments on MATH-500 and the Puzzle dataset. As a result, we adopted this 90% summation rate as a unified selection criterion throughout all experiments reported in the paper.
>
> **Sensitivity to budget selection.** We also evaluated the sensitivity of our method to different data selection budgets. The results are provided in the supplementary material (Table 2 and the table below), under the following experimental settings:
>
> - **Setting 1 (MATH-500):** We use the DeepSeek-R1-Distill-Qwen-1.5B model and evaluate Pass@1 accuracy on the MATH-500 benchmark.
> - **Setting 2 (GSM8K):** We use Qwen-2.5-1.5B-Instruct as the base model and evaluate Pass@1 accuracy on the GSM8K validation set.
> - **Setting 3 (Puzzle dataset):** We follow the Logic-R framework to generate 5,000 puzzle-style questions varying in difficulty and number of roles (2–5 roles). We then test generalization on 6–8 role puzzles using the same Qwen-2.5-1.5B-Instruct model.
>
> Across all three settings, we consistently observed that setting the summation threshold to **90%** yielded the **best trade-off between performance and data size**, and generalized well across datasets and tasks. This empirical consistency motivated our use of the 90% threshold in all main results reported in the paper.
>
>
>
> |                     |   Setting-1   |                |                |   Setting-2   |                |                |   Setting-3   |                |                |
> |---------------------|----------------|----------------|----------------|----------------|----------------|----------------|----------------|----------------|----------------
> |                     | Performance ↑  | budget δ ↓     | data efficiency ↑ | Performance ↑  | budget δ ↓     | data efficiency ↑ | Performance ↑  | budget δ ↓     | data efficiency ↑ |
> | No-training         | 82.8           | 0              | -              | 80.4           | 0              | -              | 0.1            | 0              | -              |
> | Full-data-training   | 86.2           | 40300          | 0.008%         | 94.2           | 7500           | 0.18%          | 82.7           | 5000           | 1.65%          |
> | Discard the negative | 87.9           | 39485          | 0.012%         | 96.8           | 7260           | 0.22%          | **83.5**       | 4832           | 1.72%          |
> | summation-rate = 95% | **89.2**      | 19295          | 0.033%         | **97.8**       | 6721           | 0.26%          | 83.3           | 3933           | 2.11%          |
> | summation-rate = 90% | 89.0          | 8530           | **0.072%**     | 97.4           | 1329           | **1.28%**      | 83.4           | 3805           | **2.18%**      |
> | summation-rate = 80% | 87.2          | 7024           | 0.062%         | 95.8           | 1230           | 1.25%          | 77.2           | 3619           | 2.13%          |
> | summation-rate = 70% | 80.5          | 6575           | -0.034%        | 94.3           | 1132           | 1.23%          | 64.8           | 3133           | 1.94%          |
>
>
> ---
>
>
> ### **Q2. What are the characteristics of samples that receive very high positive scores versus those that receive negative scores?**
>
> That's an excellent question! We found that after sorting the data in descending order based on the RFT-Score, the lower-ranked instances (indicating a negative impact) are primarily those where the model is consistently either completely correct or completely incorrect across all eight responses. In contrast, the higher-ranked entries are those where the model occasionally answers correctly but also makes mistakes.
>
> This underscores the importance of including questions that the model can occasionally answer correctly during the RFT process. By reinforcing learning on such data, the model can thoroughly explore potential reasoning paths, both correct and incorrect. Then, the feedback from rewards helps the model quickly identify and learn these correct reasoning abilities! We showcase several such data cases on Pages 3 and 4 of the original supporting materials.

---

> ### Author Response · Authors · 2025-08-09
> **To Dear Reviewer wKRB**
>
> Dear Reviewer wKRB,
>
> We sincerely thank you for your participation throughout the entire review cycle and for your valuable contributions to the academic community.
>
> During the rebuttal period, we have made our best efforts to address your concerns, including conducting additional experiments on new data combinations and evaluating our method on benchmarks such as GPQA and Arena-Hard. We hope that these efforts have adequately addressed your questions and clarified the strengths of our work.
>
> If possible, we would be deeply grateful if you could consider updating your evaluation to a higher score based on our responses and the new results.
>
> Thank you again for your time and thoughtful feedback.
>
> Sincerely,
>
> The Authors

---

### Official Review · Reviewer_s3Rt · 2025-07-02

**Clarity:** 3
**Significance:** 3
**Originality:** 2
**Rating:** 5
**Confidence:** 3

**Summary:**

This paper introduces RFT-DA, a data selection framework for reinforcement fine-tuning (RFT) of large language models that identifies valuable training examples by measuring how their removal affects final training rewards. The core innovation is a sample-level ablation analysis that quantifies individual training example importance through their contribution to model learning.

**Main Contributions:**
- **Data ablation scoring framework**: Defines D(z) as the difference in expected reward when sample z is removed from training, providing a direct measure of sample importance for RFT
- **Efficient first-order approximation**: Introduces a gradient-based estimator D̂(z) that measures alignment between individual sample gradients and overall training direction, achieving linear computational complexity O(NE + NC)
- **Theoretical analysis**: Provides worst-case error bounds for the approximation under Lipschitz continuity assumptions, with bounds tighter than many supervised learning influence functions
- **Empirical validation**: Demonstrates that using only 20% of data selected by RFT-DA scores achieves superior performance compared to full datasets, with trained 7B models matching o1-mini performance

The method backtracts through the optimization process using temporal differentiation and first-order Taylor approximation, resulting in a lightweight estimator that evaluates gradient alignment across training stages.

**Questions:**

1. **How does per-sample gradient computation align with modern batch-based training practices?** The method requires individual sample gradients, but modern RFT typically uses batched updates - how do you reconcile this with practical training pipelines that rely on batch statistics?

2. **Can you provide detailed comparison conditions for the claimed o1-mini performance?** The assertion that a 7B model matches or out-performs o1-mini performance across benchmarks is remarkable, but the tables don't seem to support it. i recommend some revision of this claim.

3. **Why is the separate surrogate training phase necessary rather than computing scores during actual training?** The method requires expensive surrogate training to generate checkpoints for gradient computation, but provides no ablation on whether this two-phase approach is essential - is there possible alternatives?

**Ethical Concerns:**

["NO or VERY MINOR ethics concerns only"]

**Final Justification:**

my concerns are addressed by reviewer response, thus updating final rating.

**Quality:**

3

**Strengths And Weaknesses:**

### Strengths

- **Addresses a significant gap in RFT data selection methodology**. The paper tackles an important but underexplored problem of principled data selection for reinforcement fine-tuning, moving beyond heuristic approaches to provide a theoretically grounded framework for quantifying sample importance.

- **Strong empirical validation with comprehensive benchmarks**. The evaluation spans multiple mathematical reasoning benchmarks (AIME24, MATH-500, AMC23, Minerva, OlympiadBench) across different model sizes and RL algorithms, demonstrating consistent improvements with substantial margins over existing methods (9.1 points over difficulty-based, 6.3 over LIMR).

- **Practical efficiency with theoretical guarantees**. The method achieves remarkable data efficiency (20% of data for superior performance) while providing formal error bounds for the first-order approximation, making it both practically valuable and theoretically sound for large-scale applications.

- **Good positioning of the work** in the rapidly evolving field. Well written over-all.

### Weaknesses

- **Limited theoretical justification for the gradient alignment assumption**. The core assumption that gradient alignment with the overall training direction indicates sample importance lacks rigorous theoretical foundation. The connection between local gradient similarity and global contribution to final rewards is not well-established, particularly in the complex optimization landscape of reinforcement learning.

- **Narrow domain evaluation despite RFT's broader applications**. While mathematical reasoning is a primary RFT target, the evaluation lacks validation on other important domains where RFT is commonly applied, such as scientific reasoning (GPQA), conversational AI (ArenaHard), or code generation tasks, limiting confidence in the method's general applicability beyond mathematics.

- ** Separate surrogate training is an expensive overhead**

---

> ### Author Rebuttal · Authors · 2025-07-31
>
> Response to Reviewer s3Rt:
>
>
> We sincerely appreciate your time and feedback! We hope our response addresses your concerns. If you have any further questions, please don't hesitate to let us know. We are more than happy to discuss further.
>
> ---
>
> ### **1. On the theoretical justification for the gradient alignment.**
>
>
> Great question! The use of gradient alignment, as formulated in Equation 5, is supported by both theoretical justification and empirical evidence.
>
> First, as detailed in Section 3 of the supplementary material, Equation 5 provides a **first-order approximation** of the **leave-one-out influence** of a sample z on the RFT process. This influence, formally defined in Equation 4, quantifies how the removal of a specific sample affects the training outcome, thus enabling us to assess the importance of individual samples.
>
> We further analyze the estimation error of the gradient alignment score in **Proposition 1**, where we show that our approximation of leave-one-out influence is **more accurate** than many existing influence estimation methods in supervised learning literature. This theoretical result underpins the validity of using gradient alignment as a proxy for sample importance.
>
> Intuitively, in RFT, a sample whose gradient is well-aligned with the aggregate update direction (i.e., the average gradient of other samples) is more representative and contributes more constructively to training. This idea is not only theoretically motivated but also strongly supported by our empirical results. As shown in Tables 1, 2, and 3, our approach consistently outperforms baseline methods, including those using the full dataset—demonstrating **notable performance gains** with significantly reduced training data.
>
> In summary, both the **theoretical formulation** and the **experimental validation** provide solid support for our gradient alignment assumption as a meaningful and effective criterion for sample selection in reinforcement fine-tuning.
>
> ---
>
> ### **2. More evaluation results on GPQA and ArenaHard.**
>
> During the rebuttal period, we conducted experiments on GPQA and ArenaHard. The model used for the experiments was DeepSeek-R1-Distill-Qwen-7B, and the datasets involved included DeepScaleR and a 50K subset of the Tulu-3-Preference dataset. The RL algorithm employed was Reinforce++, and the reward model used was Tulu-3-8B-RM. To ensure fairness, all methods were filtered to include only the top 90% based on summation rate.
>
> Notably, RFT-DA continues to maintain its leading advantage! We will update these results in the revision.
>
>
> Method | GPQA | Arena-Hard
> ---|---|---
> No training| 48.4 | 17.3
> HVS|52.8|26.9
> LIMR|51.4|24.2
> RFT-DA (ours)|**53.5**|**28.5**
> Full-set training|51.7|27.4
>
> Moreover, for **larger model settings**, we present the training results on the 32B (DeepSeek-R1-Distill-Qwen-32B) and 70B (DeepSeek-R1-Distill-Llama-70B) models. Here, we directly apply the subsets chosen in the 1.5B model experiments for RFT on these larger-scale models. The performance advantages of RFT-DA remain consistent with those observed on the 1.5B and 7B models, outperforming both HVS and LIMR, as well as the training results on the full set!
>
> Method|Math-500 Pass@1 for 32B|Math-500 Pass@1 for 70B
> ---|---|---
> No training|94.3|94.5
> HVS|93.7|94.1
> LIMR|92.9|93.2
> RFT-DA (ours)|**96.9**|**97.1**
> Full-set Training|95.1|96.5
>
> ---
>
> ### **3. Comparison with O1-mini.**
>
>
> The results presented in Table 1 indicate that, at the 7B model scale, our method achieves performance comparable to that of o1-mini on AMC23 and Olympiad (89.4 vs. 90.0 and 65.4 vs. 67.2). Notably, on AMC23, our method even surpasses the performance of o1-mini (91.0 vs. 90.0). We will update our statement to reflect that our method can match or even exceed the performance of o1-mini on certain benchmarks.
>
> ---
>
> ### **4. On the necessity of a separate surrogate training and scoring phase.**
>
> This is a great question! We appreciate your suggestion! We will include this new version with improved speed in our revision. The separate surrogate training and scoring phases are the main contributors to the time consumption in our method when compared to HVS and LIMR. Following your suggestion, we have also implemented a dynamic version of conducting scoring during surrogate training as below.
>
>
>
> **RFT-DA Dynamic: A Faster and More Efficient Variant:** To further reduce computation time for scoring, we introduce **RFT-DA Dynamic**, which eliminates the need for fixed model checkpoints by computing scores dynamically during surrogate training. Specifically, at iteration $t$, given a batch $B_t$ and a sample $z \in B_t$, we compute and store the inner product between the gradient of $z$ and the average gradient of the batch $B_t$. After training, we aggregate these stored inner products across all iterations in which each sample was involved to estimate its final score:
>
> $$D(z) \approx \sum_{t \in \{t^z_1, t^z_2, \ldots\}} \frac{2\eta_t}{N} \langle G_z^t, G_{B_t}^t \rangle,$$
>
> where $t^z_i$ denotes the iteration index of $i$-th time in which sample $z$ was used for training. This approach allows RFT-DA Dynamic to accumulate all scoring information on-the-fly during training, thus incurring negligible additional computation.
>
> The table below summarizes the comparison across methods in terms of scoring time using 8 GPUs and performance:
>
>
>
> Method   | $\~$  Time Cost of Scoring   |  $\~$ Math-500 Pass@1
> ---|---|---
> HVS| 4.7 sec $\approx 0$|83.3
> LIMR| 5.3 sec $\approx 0$ |83.6
> RFT-DA (ours)|63.1 Hours| **89.0**
> RFT-DA-fast (ours)|5 Hours| **88.1**
> RFT-DA-Dynamic (ours) | 32 sec | **88.3**
> Full-set training | - | 86.2
>
> **RFT-DA Dynamic** maintains comparable performance as RFT-DA while being much more efficient, making it a highly effective and scalable solution for data selection under budget constraints.

---

> > ### Comment · Reviewer_s3Rt · 2025-08-04
> >
> > Thanks to authur response.
> >
> > My concerns are largely addressed

---

> > > ### Author Response · Authors · 2025-08-04
> > > **Thanks for the response**
> > >
> > > We appreciate your response! Should you have any further inquiries, please don’t hesitate to get in touch. We are eager to assist with any concerns you may have.

---

### Official Review · Reviewer_L7kr · 2025-07-02

**Clarity:** 2
**Significance:** 3
**Originality:** 3
**Rating:** 4
**Confidence:** 3

**Summary:**

This paper presents data ablation for reinforcement fine-tuning (RFT-DA), a framework that can quantify the importance of individual training examples in RFT. RFT-DA uses an efficient first-order approximation approach to estimate the importance of a sample by evaluating the alignment between its gradient and the gradient over all samples. The samples with the highest alignment scores are kept for RFT training. Experimental results demonstrate the effectiveness of this method.

**Questions:**

- The inclusion of the “Motivation” and “Our Approach” paragraph titles in the Introduction section seems to be unnecessary.
- Line 121: It is inappropriate to equate a set with a sample
- Line 168: The function $\mathcal{J}(\cdot)$ should denote the expected advantage rather than the expected reward, consistent with the definition in Equation 1.
- Line 180: There is no $\mathcal{Z} / z$ in Equation 5.
- Line 181: There is no $T$ in Equation 5.
- It may be more appropriate to relocate Figure 1 and its accompanying description to the Experiments section, as it illustrates empirical behavior rather than methodological design.
- Line 265: Broken section reference.
- Line 295: Should be $1e-6$ rather than $1\times e^{-6}$.
- Line 387: “good enough” is too informal for the academic context.
- Line 449: Wrong authorship of Olympiadbench.
- Supplementary Material Line 82: Should be $\mathcal{J}^0(z)=0$.
- How is RFT-DA compared to LIMR in terms of computational efficiency?

**Ethical Concerns:**

["NO or VERY MINOR ethics concerns only"]

**Final Justification:**

Most of my concerns have been addressed, except for the issue of time complexity. The presentation issues cannot be fully resolved at the discussion stage. Since this paper offers a clear contribution despite its weaker presentation, I rate it as a borderline accept.

**Limitations:**

Yes

**Paper Formatting Concerns:**

There are no paper formatting concerns.

**Quality:**

3

**Strengths And Weaknesses:**

- Strengths
    - The paper presents a clear and well-motivated research objective.
    - The proposed method is straightforward and achieves strong empirical performance.
- Weaknesses
    - The analysis of computational complexity lacks rigor.
        - The authors claim that the total computational complexity for scoring the entire dataset is $O(NE + NC)$, but the definitions of $E$ and $C$ are unclear. From my understanding, both seem to refer to the number of training epochs. If that is the case, the notation may be misleading. The authors may mean to use $NE$ to represent the computational cost of training the model for $E$ epochs and saving checkpoints, and $NC$ to denote the cost of computing the data ablation score for each sample. However, these two components represent different stages of the pipeline and should not be directly summed.
    - The overall presentation of the paper could be improved. See questions.

---

> ### Author Rebuttal · Authors · 2025-07-31
>
> Response to Reviewer L7kr's Comments:
>
> ---
>
> We sincerely thank the reviewer for the insightful feedback. We appreciate the constructive criticism and have carefully addressed the concerns raised regarding the computational complexity analysis and the overall presentation of the paper.
>
> ---
>
> ### **1. On the computational complexity analysis.**
>
> We will revise this! Regarding the definitions of NE, N, and C, please refer to Line 212 of the manuscript. Following your suggestion, we will clarify the definitions of N, C, and E to improve clarity.
>
> Specifically:
>
> - \( N \) denotes the total number of training samples,
> - \( E \) is the number of training epochs,
> - \( C \) is the number of checkpoints selected for score computation.
>
>
> Accordingly, the computational complexity of the surrogate training phase is \( O(NE) \), while the scoring process incurs a complexity of \( O(NC) \). Therefore, the overall computational complexity is \( O(NE) + O(NC) \).
>
> ---
>
>
> ### **2. On the overall presentation:**
>
> We appreciate the feedback. We have carefully revised the manuscript to improve clarity, flow, and readability. Specifically:
>
> (a). Line 121: This was a typo caused by a compilation error. We have corrected it: the correct expression should be "Z = {z_i = (x_i, y_i)|_{i=1}^N }". The {} notation is not properly rendered in math mode.
>
> (b). The function $J(\cdot)$ denotes the expected advantage, and we will ensure it remains consistent with the definition provided in Equation (1).
>
> (c).Equation (5) indeed does not involve $Z \setminus z$, and we will remove the reference to $Z \setminus z$ in Line 180. Additionally, we will explicitly include the time horizon $T$ in the summation notation for clarity.
>
> (d).The incorrect section reference in Line 265 was due to a compilation error, and we will correct it in the revised version. And we have updated the reference to OlympiadBench: Chaoqun He, Renjie Luo, Yuzhuo Bai, et al., OlympiadBench, ACL 2024.
>
> (e).Line 82 in the Supplementary Material should be $D^0(z) = 0$. Thank you for catching this typo, and we will correct it accordingly.
>
>
> ---
>
>
> ### **3. Speed Comparison with LIMR:**
>
> In the supplementary material (Table 3), we include a comparison with HVS on the MATH-500 dataset. Notably, HVS incurs virtually no additional time cost during the scoring phase. LIMR demonstrates comparable computational cost, as both rely solely on reward statistics collected during training and do not require any additional computation at inference. However, both methods exhibit significantly lower performance (Pass@1 of 83.6) compared to full-set training (86.2), limiting their practical utility in performance-critical scenarios.
>
> In contrast, our proposed approach achieves substantially higher performance: 89.0 with RFT-DA and 88.3 with the faster variant, RFT-DA Fast. Despite these gains, the total computation time—including surrogate training, scoring, and subset training—is significantly lower than full-set training (192.2 hours for RFT-DA, 134.2 hours for RFT-DA-Fast vs. 277.6 hours for full-set training).
>
> These results will be added to the main paper for clarity. It’s worth noting that the scoring time is measured using 8 GPUs, which can be significantly reduced by using more GPUs.
>
>
> ---
>
> **RFT-DA Dynamic: A Faster and More Efficient Variant:**
>
> To further reduce computation time for scoring, we introduce **RFT-DA Dynamic**, which eliminates the need for fixed model checkpoints by computing scores dynamically during surrogate training. Specifically, at iteration $t$, given a batch $B_t$ and a sample $z \in B_t$, we compute and store the inner product between the gradient of $z$ and the average gradient of the batch $B_t$. After training, we aggregate these stored inner products across all iterations in which each sample was involved to estimate its final score:
>
> $$D(z) \approx \sum_{t \in \{t^z_1, t^z_2, \ldots\}} \frac{2\eta_t}{N} \langle G_z^t, G_{B_t}^t \rangle,$$
>
> where $t^z_i$ denotes the iteration index of $i$-th time in which sample $z$ was used for training. This approach allows RFT-DA Dynamic to accumulate all scoring information on-the-fly during training, thus incurring negligible additional computation.
>
> The table below summarizes the comparison across methods in terms of scoring time using 8 GPUs and performance:
>
>
>
>
> Method   | $\~$  Time Cost of Scoring   |  $\~$ Math-500 Pass@1
> ---|---|---
> HVS| 4.7 sec $\approx 0$|83.3
> LIMR| 5.3 sec $\approx 0$ |83.6
> RFT-DA (ours)|63.1 Hours| 89.0
> RFT-DA-Dynamic (ours) | 32 sec | 88.3
>
> While **RFT-DA-Dynamic** yields slightly lower accuracy than the original RFT-DA, it achieves comparable computational efficiency to HVS and LIMR, requiring less than one minute for scoring. Importantly, it still significantly outperforms HVS, LIMR, and full-set training in terms of final model performance, making it a highly effective and scalable solution for data selection under budget constraints.

---

> > ### Comment · Reviewer_L7kr · 2025-08-06
> >
> > Thank you for your response.
> >
> > For the complexity analysis, I still think that it's wrong to add $NE$ and $NC$, neither in the form of $O(NE+NC)$ nor $O(NE)+O(NC)$. $NE$ and $NC$ have different units. It makes no sense to add them up. My other concerns have been addressed. I will keep my score.

---

> > > ### Author Response · Authors · 2025-08-08
> > > **Thanks for your response.**
> > >
> > > Thank you for your participation in the discussion phase, as well as for your contributions to the community and your recognition of our article!
> > >
> > > We would like to explain why the overall complexity is $O(NE)+O(NC)$. If you have any other viewpoints, we would greatly appreciate your feedback!
> > >
> > > In fact, the computational complexity of neural networks is proportional to the amount of data and the number of computations performed. Clearly, during surrogate training, computations are carried out on N data points, with each data point undergoing E computations (where E is the number of epochs). Therefore, the complexity during the training phase is proportional to EN.
> > >
> > > In the scoring phase, we select C checkpoints, and after loading each checkpoint, we need to process the entire dataset. Thus, the complexity for scoring is proportional to CN. Additionally, our algorithm requires inference and gradient calculation during both the surrogate training and scoring phases. This is why we combine the two, resulting in an overall complexity of  $O(NE)+O(NC)$.

---

### Note · Authors · 2025-08-12

Dear AC:

We thank you and all reviewers for the time. We summarize the key points below.

Reviewer | Strength| Questions | Initial rate | Our reply | Feedback
---|---|---|---|---|---
L7kr | clear and well-motivated, straightforward and with strong empirical performance| **A.** The complexities of each part cannot be added together. **B.** Some typos. **C.** Efficiency compasison with LIMR | 4: BA | **1**. We explain why all part complexities can be added together (see https://openreview.net/forum?id=mjhCFB3HLQ&noteId=vsJfUfg0P8) **2**. Fix typos.  **3**. We present a faster version: **RFT-DA-Dynamic**, which shows exceptional speed and significantly outperforms others. | Overall satisfied
 | | | | |
s3Rt| Fills an RFT data-selection gap with strong empirical results, efficient methods backed by theoretical guarantees, and good positioning.| **A.** Want to see the theoretical justification. **B.** More evaluation on GPQA and Arena. **C.**. Want to see the settings of the comparison with O1-mini **D.** Suggest to merge the originally separate surrogate training and scoring phases.  | 4: BA | **1.** We provide a thorough restatement of the existing theoretical and empirical justifications presented in the article. **2.** We provide test results on relevant benchmarks (leading in performance). **3.** We further clarify some details in comparison with O1-mini. **4.** We present a faster version: **RFT-DA-Dynamic**, which shows exceptional speed and significantly outperforms others. | Overall satisfied.
 | | | | |
wKRB| Clear and readable; novel and well-justified; the performance is generalizable.|**A.** Evaluation on more benchmarks and larger models. **B.** Selection and sensitivity of budget.  | 4: BA | We provide more experiments on more benchmarks and larger models, also with the sensitivity study of the budget setting (already in the supp material)| Without Reply
 | | | | |
rGmp | Well-written; novel with clear theory and strong experiments.| **A.** On the computation cost **B.** On the Selection and Sensitivity of the Budget. | 4: BA | **1.** We clarify that RFT-DA incurs minimal time costs compared to full dataset training, while the selected subset significantly outperforms it and other methods. Furthermore, we present a faster version: **RFT-DA-Dynamic**, which shows exceptional speed and significantly outperforms others. **2.** We provide the sensitivity study of the budget (already in the supp material)| Concerns resolved

---

### Decision · Program_Chairs · 2025-09-17

**Decision:**

Accept (poster)

**Comment:**

This paper studies a timely problem, data selection for RFT. This paper gives an intuitive method with extensive and strong empirical results. All reviewers like the paper. The AC agrees and recommends acceptance.